# As time goes by: Long-term retention of economics skills

**Douglas McKee**[ID]◉*, **George Orlov**[ID]◉

Department of Economics, Cornell University, Ithaca, New York, United States of America

◉ These authors contributed equally to this work.
* douglas.mckee@cornell.edu

## Abstract

The vast majority of research on student learning is based on assessments of student knowledge given during or at the end of an academic term. Until now, we have known very little about what knowledge students retain after a course is over or what determines how much they retain. In this paper, we analyze data collected from students who took one of six courses in introductory or intermediate microeconomics. All students in these courses took a low-stakes standard assessment of their learning at the end of the term. At follow-ups, one to 2.5 years later, these students were surveyed about their academic and job-related activities, and given the same assessment they took at the end of the course. We find that some demographic characteristics and prior preparation for the course are strong predictors of how much students retain while initial attitudes toward economics are not. We also find evidence that for some students, application of economic skills in subsequent jobs and courses helps students retain course skills.

## Introduction

As exemplified by the meta-analysis of [1], there is ample evidence supporting the influence of pedagogical methods on student achievement. However, the overwhelming majority of research on student learning focuses on the immediate gains rather than the long-term effects on student outcomes. While there is a sizable literature analyzing students' retention (i.e., continuation of study) in a subject or major (e.g., [2]), very few studies have looked at how well students retain the knowledge and skills that they have been taught as time passes. Our project's goal is to fill this gap in the literature.

Although psychology research on short-term memory considers retention intervals of seconds, and research on long-term memory focuses on retention measured in minutes to a few weeks, educators hope their students will remember what they have learned for months or years. In this paper, we refer to this very long-term memory as *learning retention*. The literature on learning retention is extremely limited,

**Data availability statement:** All data used in this study are available at: https://doi.org/10.5281/zenodo.17517274.

**Funding:** This paper is based upon work supported by the National Science Foundation under Grant 2021094, and any opinions, findings, and conclusions or recommendations expressed in this paper are those of the authors and do not necessarily reflect the views of the National Science Foundation.

**Competing interests:** The authors have declared that no competing interests exist.

because it requires longitudinal data on students, which in the context of higher education can be difficult and expensive to collect. It is also important for samples to be large enough for statistical inference and representative of the student population of interest.

[3] used a sample of 92 students who were given a set of questions originally asked on their consumer behavior course final exam as an extra credit test in a marketing course (usually taken after consumer behavior). The data were collected over three years. For this sample, retention intervals ranged from 8 to 101 weeks (with 12 unique intervals). The authors argued that students performed better on items that tested recognition of concepts in a scenario or identification of scenarios associated with a concept when compared to items that required recall of definitions. Furthermore, concepts on which the students were tested more than once had better retention rates.

[4] measured the changes in the Force Concept Inventory (FCI) to assess student learning retention in physics courses. They observed little decline in the FCI scores over several years; however, the study did not demonstrate whether the sample of 127 students who responded for the retention study was representative of the target student population.

[5] examined the longitudinal effect of pedagogical transformations in introductory physics courses at the University of Colorado using the Brief Electricity & Magnetism Assessment (BEMA). While learning retention is not the focus of the paper, there is a brief section on tracked students who took BEMA at the end of the freshman physics course and again at the start of the junior Electricity & Magnetism course. The author found that these students scored, on average, only approximately 5% lower on the post-test (administered junior year) compared to their BEMA scores in freshman physics. The author highlights that the tracked students were all physics majors and represent a highly selective sample.

[6] used the Quantum Mechanics Conceptual Survey (QMCS) to study learning retention. Comparing two similar cohorts of students, they demonstrated that while the decline in the percentage of correct answers for a post-test administered 6 months after the pre-test is small to begin with, the retention of material was higher in the cohort taught using interactive engagement methods.

[7] measured retention using a test that consists of three open-ended questions testing one concept from Newtonian Physics and is administered, without a baseline test, one and two years after the students complete their freshman Principles of Physics course. The authors claimed that understanding of the material improved over time, potentially due to coursework performed in the time between the testing, but there is no quantitative evidence of this effect.

Finally, [8] study students' learning and knowledge retention in two introductory economics courses taught over multiple academic terms using various teaching methods. Students are given the same set of 10 multiple choice questions on the first day of class (before the material is taught), on the ninth day of class (after the material is taught), and on the last day of class. They compare the second test to the first test to measure initial learning, and use the difference between the third test and the second test as a measure of knowledge retention. They find that on average, student

knowledge declines between the second test and the end of the term, but they say nothing about retention beyond this point.

One limitation shared by all these studies is that none empirically examined the potential mediating role of relevant coursework completed, or work experience acquired within the retention interval, and only [8] considers the role of student characteristics. These factors might hold a great deal of information that could be useful for educators in designing their course plans. That is, if some groups systematically retain less of what we teach, instructors could pay particular attention to those groups to instill a deeper level of understanding. Similarly, instructors should consider encouraging post-course behavior associated with higher retention of course skills. Our paper addresses this important limitation.

Another key innovation of our study is that we collect data from a large sample of students in introductory and intermediate-level courses spread across three semesters and follow them over several years. These students were given standard assessments that evaluated their skill in the course material at the end of each of these courses. One to 2.5 years later, we followed up with these students, re-administered the assessment of economics skills for the appropriate course, and asked them about their professional and academic behavior in the interim period.

## Methods

In this study, we collect demographics and performance on assessments for two courses (Introductory Microeconomics and Intermediate Microeconomics) offered at Cornell University in three semesters each between Spring 2019 and Fall 2022. We follow up with students who took these courses over the following one to 2.5 years and give them performance assessments and surveys on their academic and professional activities during the interim period. We use regression models to investigate how initial performance, demographic characteristics, attitudes toward economics, and interim behavior predict changes in the performance measures over time.

Note that the change in assessment score is not a perfect measure of knowledge retention. For example, a student's score could remain the same at the follow-up if they forget some course concepts they knew at the end of the course but also learn course concepts they had not learned during the course. In this case, the change in score would over-estimate the amount of knowledge that is actually retained by the student.

While the instructor and content of all three cohorts of each course were the same, there were differences in how these courses were taught. Specifically, some courses were taught online during the COVID-19 pandemic, while others were taught in-person before or after the pandemic. Similarly, the instructors spent different fractions of class time lecturing and having students engage in problem solving activities in the different academic terms. We estimate our regression models using pooled data from all the cohorts for a given course, allowing us to maximize our sample size while still including cohort fixed effects to account for unobserved differences between cohorts. That is, our models allow the average change in assessment scores to differ by cohort, holding student characteristics constant.

## Methods: Introductory microeconomics

Fig 1 shows an overview of the data collection process that started in Fall 2019 and continued until Fall 2024. The course we follow was taught in Fall 2019, Fall 2020, and Fall 2022. As noted above, the learning goals in all three academic terms were the same, and in all three terms the course was taught by the same instructor. The course follows a traditional textbook (Krugman and Wells, 5th edition) and uses graphs and equations to define core microeconomic concepts including supply and demand, firm behavior, market structure, equilibrium, and externalities, and use them to make predictions about real world outcomes. Students are assigned 20 online problem sets during the term, and exams are closed book with a mix of multiple choice and short answer questions. The amount of active learning used in the classroom varied across the semesters as did the course modality (in-person versus online). We explore the impact of these differences in a separate paper.

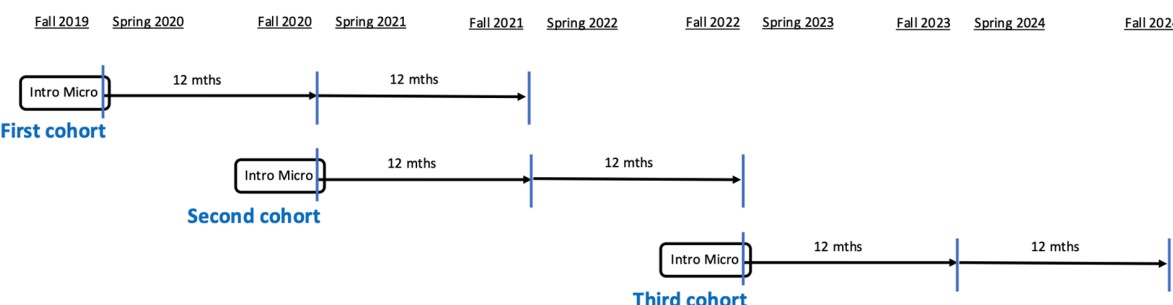

**Fig 1. Introductory microeconomics timeline.** The data collection process for students who take introductory microeconomics

Students in these three courses were invited to complete online surveys and assessments for the study on the first day of the term (August 29, 2019; September 2, 2020; and August 22, 2022). Participants were presented with an information statement describing the study's purpose, procedures, and their rights, including the voluntary nature of participation. Informed consent was obtained by having participants click a button indicating their consent before beginning the survey. Non-consenting students and students under age 18 were not included in the study. All data was deidentified before it was analyzed.

At the end of each term when the course was taught, students were given the Principles of Economics Skills Assessment for Microeconomics (PESA-Micro). This test is a part of the Cornell Suite of Economic Skills Assessments that is publicly available to economics instructors at econ-assessments.org [9], and it is composed of 30 multiple choice questions that evaluate student knowledge of the learning goals in a typical introductory microeconomics course. While no assessment can be a complete measure of learning in a course, economics and other STEM disciplines have a strong tradition of using well-designed multiple choice tests as proxies for course learning. These include the Test of Understanding College Economics [10], the Force Concept Inventory [11], the Genetics Concept Assessment [12], and the Chemical Concept Inventory [13]. PESA-Micro was developed using the best practices outlined in [14]. Each assessment question in PESA-Micro maps to specific learning goals of the relevant course to ensure broad coverage, and the questions were developed using an iterative process that included student think-aloud interviews, faculty feedback, and pilot data. PESA-Micro scores are strongly correlated with final exam scores of students in each of the the three cohort samples of introductory microeconomics with correlation coefficients of 0.53, 0.45, and 0.20.

PESA-Micro was administered online, and students could earn a small amount of extra credit for taking the test. Their score on the test did not affect their grade in the course, but students were advised that putting in a good effort would help them understand where they needed to focus their studying for the final exam and would help the instructor improve the teaching of the course in the future. These students were also given the Math for Economics Skills Assessment, Foundations (MESA-Foundations) at the beginning of the term to measure their mathematical preparation for the course.

The MESA-Foundations score is a critical control in the models we present below, and the primary goal of our study is to explain changes in PESA-Micro scores between the end of the term in future follow-ups. For these reasons, we define our target sample for each cohort to be every student that took PESA-Micro and MESA-Foundations at the end of their course and consented to be part of the research study. While these target samples are subsets of the broader groups of students enrolled in each course, Table 1 shows that these samples are quite representative of the broader groups. Average scores on final exams were not significantly different for the Fall 2019 cohort and 1.4-2.2 points higher for test-takers relative to the overall average in the other two cohorts. Freshmen were significantly more likely to participate in Fall 2020 and Fall 2022, and sophomores in these two cohorts were significantly less likely to participate.

**Table 1**. **Selectivity of target samples.**

| Cohort | All UG Students | Target Sample |
|---|---|---|
| Introductory Microeconomics, Fall 2019 | | |
| Final Exam | 73.53 | 74.09 |
| Freshman (fraction) | 0.51 | 0.52 |
| Sophomore (fraction) | 0.37 | 0.35 |
| Junior (fraction) | 0.10 | 0.11 |
| Senior (fraction) | 0.03 | 0.02 |
| N | 440 | 184 |
| Introductory Microeconomics, Fall 2020 | | |
| Final Exam | 83.75 | 85.16** |
| Freshman (fraction) | 0.70 | 0.74** |
| Sophomore (fraction) | 0.18 | 0.15** |
| Junior (fraction) | 0.07 | 0.07 |
| Senior (fraction) | 0.05 | 0.04 |
| N | 458 | 315 |
| Introductory Microeconomics, Fall 2022 | | |
| Final Exam | 87.22 | 89.39** |
| Freshman (fraction) | 0.38 | 0.48** |
| Sophomore (fraction) | 0.51 | 0.42** |
| Junior (fraction) | 0.09 | 0.08 |
| Senior (fraction) | 0.03 | 0.02 |
| N | 441 | 202 |
| Intermediate Microeconomics, Spring 2019 | | |
| Final Exam | 71.79 | 72.77 |
| Freshman (fraction) | 0.39 | 0.39 |
| Sophomore (fraction) | 0.39 | 0.38 |
| Junior (fraction) | 0.13 | 0.13 |
| Senior (fraction) | 0.09 | 0.10 |
| N | 75 | 61 |
| Intermediate Microeconomics, Fall 2019 | | |
| Final Exam | 78.76 | 80.60** |
| Freshman (fraction) | 0.06 | 0.07 |
| Sophomore (fraction) | 0.59 | 0.60 |
| Junior (fraction) | 0.27 | 0.26 |
| Senior (fraction) | 0.08 | 0.07 |
| N | 183 | 140 |
| Intermediate Microeconomics, Fall 2020 | | |
| Final Exam | 85.04 | 84.62 |
| Freshman (fraction) | 0.06 | 0.04 |
| Sophomore (fraction) | 0.53 | 0.53 |
| Junior (fraction) | 0.35 | 0.36 |
| Senior (fraction) | 0.07 | 0.07 |
| N | 171 | 120 |

Symbols represent tests of equality of means for students selected for follow up and those that were not: $^+ p < 0.1$, $^* p < 0.05$, $^{**} p < 0.01$.

We use a two-stage follow-up strategy to reach out to each student in our target samples one year and two years later. In the first stage, we identified several economics courses offered in the follow-up period that contained at least three target students and where the instructor was willing to give their students extra credit in their course for participating in the study. All students in these courses were offered extra credit for participation, though we only use data from our target students for this project. Over 80% of the students in these classes chose to participate, allowing us to inexpensively reach about 10% of our target students. In the second stage, we offered the rest of our target students gift cards for participation. Participating students who were still on campus received $25 gift cards, and students who had left campus

received $50 gift cards since most of them worked full-time and were likely to be less responsive to cash incentives. Total response rates (shown in Tables 2–4) ranged from 30% to 46% at the one-year follow-up and from 25% to 34% at the two-year follow-up. We describe our follow-up samples in more detail and assess their representativeness of the target samples in the Data Description section below.

Students who provided informed consent and chose to participate in a follow-up survey took the online assessment (PESA-Micro) again and answered several questions about their behavior during the year since they completed the introductory microeconomics course. Specifically, we asked about courses they had taken that used their economics knowledge and jobs they held where they applied their economics knowledge. We use the measure of time spent on the online assessment as a proxy for student effort on the online assessment in order to check the validity of submitted answers, and we drop students who spent 10 minutes or less from our analysis sample. We followed up with students that were still enrolled in courses at Cornell as well as students who had graduated from Cornell or left for other reasons.

We model student scores on the follow-up assessment using the following regression model:

$$\text{PESA}_{i,t} = \beta_0 + \beta_1 \text{PESA}_{i,t-1} + \mathbf{X}_i^{\text{course}}\beta_2 + \mathbf{X}_i^{\text{demo}}\beta_3$$
$$+ \beta_4 \text{MESA}_{i,t-1} + \mathbf{X}_{i,t-1}^{\text{attitudes}}\beta_5 + \mathbf{X}_{i,t}^{\text{behavior}}\beta_6 + \varepsilon_{i,t} \tag{1}$$

This specification allows a student's follow-up score to depend on the student's characteristics measured at the beginning of the course (demographics, prerequisite skills), attitudes toward economics at the end of the course, academic and professional activities undertaken between the end of the course and the follow-up, and the student's score on the original end-of-term assessment of what was learned in the course. [15] and [16] recommend this specification when analyzing changes in student scores over time as well as an alternative model where the dependent variable is the difference in the scores and the earlier score is not included as an explanatory variable. We have estimated both models, and the results are very similar. Results for the alternative model are available upon request.

**Table 2. Introductory microeconomics samples: Fall 2019.**

|  | Target Sample | Followed in | |
|---|---|---|---|
|  |  | **Fall 2020** | **Fall 2021** |
| PESA (%) | 57 | 60* | 59 |
| PESA at follow-up |  | 54 | 53 |
| Freshman | 0.52 | 0.54 | 0.52 |
| Sophomore | 0.35 | 0.35 | 0.37 |
| Junior | 0.11 | 0.11 | 0.11 |
| Senior | 0.02 | 0.01 | 0 |
| Female | 0.60 | 0.65 | 0.74** |
| URM | 0.16 | 0.13 | 0.06* |
| First Gen | 0.14 | 0.12 | 0.10 |
| Econ Major | 0.24 | 0.30+ | 0.27 |
| MESA Foundations (%) | 81 | 83 | 82 |
| Found Economics Interesting | 0.48 | 0.45 | 0.50 |
| Thought about Economics Daily | 0.36 | 0.39 | 0.39 |
| Job Used Economics |  | 0.14 | 0.37 |
| Econ Course (interim or current) |  | 0.65 | 0.66 |
| Econ Course (interim) |  | 0.56 | 0.61 |
| Econ Course (current) |  | 0.43 | 0.37 |
| N | 184 | 84 | 62 |
| Response Rate |  | 0.46 | 0.34 |

Symbols represent tests of equality of means for those students who were successfully followed and those that were not: + $p < 0.1$, * $p < 0.05$, ** $p < 0.01$.

**Table 3**. Introductory microeconomics samples: Fall 2020.

| | Target Sample | Followed in | |
| --- | --- | --- | --- |
| | | Fall 2021 | Fall 2022 |
| PESA (%) | 53 | 55 | 55+ |
| PESA at follow-up | | 52 | 49 |
| Freshman | 0.74 | 0.79 | 0.71 |
| Sophomore | 0.15 | 0.15 | 0.18 |
| Junior | 0.07 | 0.05 | 0.08 |
| Senior | 0.04 | 0.02 | 0.04 |
| Female | 0.57 | 0.58 | 0.59 |
| URM | 0.18 | 0.21 | 0.22 |
| First Gen | 0.11 | 0.16+ | 0.20** |
| Econ Major | 0.19 | 0.24+ | 0.19 |
| MESA Foundations (%) | 80 | 82 | 80 |
| Found Economics Interesting | 0.50 | 0.51 | 0.51 |
| Thought about Economics Daily | 0.40 | 0.51** | 0.41 |
| Job Used Economics | | 0.15 | 0.42 |
| Econ Course (interim or current) | | 0.73 | 0.71 |
| Econ Course (interim) | | 0.62 | 0.61 |
| Econ Course (current) | | 0.56 | 0.47 |
| N | 315 | 103 | 79 |
| Response Rate | | 0.33 | 0.25 |

Symbols represent tests of equality of means for those students who were successfully followed and those that were not: + $p < 0.1$, * $p < 0.05$, ** $p < 0.01$.

**Table 4**. Introductory microeconomics samples: Fall 2022.

| | Target Sample | Followed in | |
| --- | --- | --- | --- |
| | | Fall 2023 | Fall 2024 |
| PESA (%) | 51 | 58** | 57** |
| PESA at follow-up | | 57 | 57 |
| Freshman | 0.48 | 0.49 | 0.50 |
| Sophomore | 0.42 | 0.39 | 0.36 |
| Junior | 0.08 | 0.11 | 0.14 |
| Senior | 0.02 | 0 | 0 |
| Female | 0.59 | 0.51 | 0.60 |
| URM | 0.13 | 0.16 | 0.12 |
| First Gen | 0.17 | 0.22* | 0.22 |
| Econ Major | 0.18 | 0.28* | 0.16 |
| MESA Foundations (%) | 87 | 90 | 89 |
| Found Economics Interesting | 0.39 | 0.38 | 0.40 |
| Thought about Economics Daily | 0.29 | 0.30 | 0.30 |
| Job Used Economics | | 0.16 | 0.38 |
| Econ Course (interim or current) | | 0.82 | 0.74 |
| Econ Course (interim) | | 0.77 | 0.72 |
| Econ Course (current) | | 0.56 | 0.48 |
| N | 202 | 61 | 50 |
| Response Rate | | 0.30 | 0.25 |

Symbols represent tests of equality of means for those students who were successfully followed and those that were not: + $p < 0.1$, * $p < 0.05$, ** $p < 0.01$.

$\mathbf{X}_{i,t-1}^{\text{demo}}$ is a vector of binary student demographic variables including college year, gender, under-represented minority (URM) status, first-generation college-goer status, and whether the student has declared (or plans to declare) an economics major.

All scores that enter our models are standardized using the distribution from the original cohort's baseline distribution to simplify interpretation and comparison of coefficients. That is, we subtract the mean score in the original cohort and divide by its standard deviation yielding baseline scores that are mean zero and unit variance.

$MESA_{i,t-1}$ is the standardized score on the math test given at the beginning of the term and $PESA_{i,t-1}$ is the standardized score on assessment of course skills given the at the end of the term. $PESA_{i,t}$ is the standardized score on the follow-up assessment.

$\mathbf{X}_{i,t-1}^{attitudes}$ is a vector of two binary measures of student attitudes toward economics. These variables reflect whether the student strongly agreed with the following statements:

1. "I think economics is interesting and applicable for people like me."
2. "I think about economic events I experience and witness in day to day life (e.g., in your own life and decisions, on the news, internet articles, etc.)"

$\mathbf{X}_{i,t}^{behavior}$ is a vector of variables representing work and course enrollment since the end of the course or the previous follow-up. These variables are constructed from answers to the following three yes/no questions given to participants during the follow-up survey:

1. Have you completed any courses where you extended or applied your knowledge of microeconomics since [DATE OF END OF COURSE IF FIRST FOLLOW-UP or DATE OF FIRST FOLLOW-UP IF SECOND FOLLOW-UP]?
2. Are you currently enrolled in any courses where you are extending or applying your knowledge of microeconomics?
3. Since you took introductory microeconomics, have you worked in any job where you used your microeconomics knowledge?

In some models, we use binary indicators based on answers to each of the above questions, and in some specifications we combine the two course measures into one that measures whether they were currently enrolled in a course that used their economics knowledge or if they completed at least one since the end of the course.

While all respondents who answer these questions affirmatively believe they are either applying or extending their knowledge, there is some inherent ambiguity as different respondents will interpret the questions differently. Some will only consider only cases where they make calculations similar to those made in their classes, and others will report more advanced courses or jobs where they encountered concepts from their introductory course.

## Methods: Intermediate microeconomics

The study design for the intermediate microeconomics students is very similar to the design described above. Fig 2 shows an overview of the data collection process which started in Spring 2019 with the first cohort and ended in Spring 2023 with the 2.5 year follow-up of the Fall 2020 cohort. As was the case for the introductory course, the learning goals and the instructor of the intermediate course remained the same throughout the project. This is a mathematically intensive course that uses the tools of calculus, constrained optimization, and game theory to put a formal foundation under the economic concepts taught in an introductory microeconomics course. Students complete six challenging pencil-and-paper problem sets during the term, and the three major exams are composed of multipart problems that require significant mathematical calculation and deep conceptual understanding. As was the case for the introductory microeconomics course we study here, there was significant variation in the amount of active learning students did in the classroom and in the course modality.

In each of the three academic terms, students were invited to complete online surveys and assessments for the study on the first day of the term (January 22, 2019; August 29, 2019; and September 2, 2020). These students, like those in the Introductory Microeconomics cohorts, explicitly consented to participating in the research study before completing the

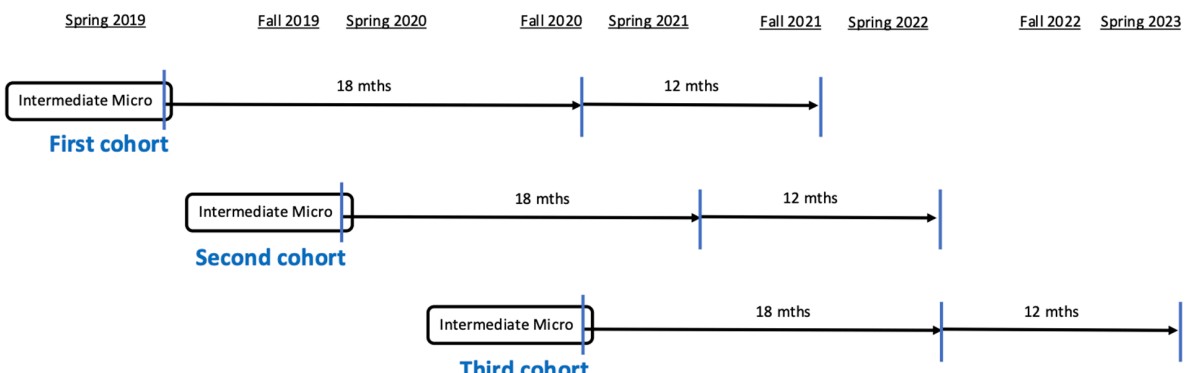

**Fig 2. Intermediate microeconomics timeline.** The data collection process for students who take intermediate microeconomics.

survey or assessment. Non-consenting students and students under age 18 were not included in the study. All data was deidentified before it was analyzed.

Students were given the Intermediate Economics Skills Assessment for Microeconomics (IESA-Micro) at the end of each of the three study courses. This test is made up of 31 multiple-choice questions that evaluate the learning goals of a typical calculus-based course in intermediate microeconomic theory. Like PESA-Micro, IESA-Micro was developed using the process described in [14], and scores are highly correlated with final exam scores in all three cohorts with correlation coefficients of 0.45, 0.56, 0.33.

It was administered in-person in Spring 2019 and Fall 2019, and online in Fall 2020. To measure their level of preparation for the course at the beginning of the term, students were given the Math for Economics Skills Assessment, Intermediate (MESA-Intermediate) and PESA-Micro, the same test that the introductory students took at the end of the term.

Table 1 shows that the between 70% and 81% of students enrolled in these courses completed the both pre-assessments and IESA-Micro assessment at the end of the term. These students (i.e., our target samples) were very similar to those students who did not take the assessments. We observe a small (1.8 percentage point) significant difference in the final exam scores between target sample and the other students only in Fall 2019.

We used the same two-stage follow-up strategy described above to reach out to our target samples for Intermediate Micro. This resulted in surveying and assessing 38-48% of students in each cohort at the 1.5-year follow-up and 25-31% of the target students at the 2.5-year follow-up.

Students that were successfully followed were given the same IESA-Micro assessment again and surveyed about economics-related courses they may have taken since the end of the course and any jobs they had where they applied the economic ideas they had learned in their courses. We also drop students from the analysis sample who spent 10 minutes or less on the assessment, and we followed both students that were still taking courses at Cornell and those that had left campus.

The regression models we use to study the determinants of skills retention are very similar to those described above:

$$\text{IESA}_{i,t} = \beta_0 + \beta_1 \text{IESA}_{i,t-1} + \mathbf{X}_i^{\text{course}}\beta_2 + \mathbf{X}_i^{\text{demo}}\beta_3 + \beta_4 \text{MESA}_{i,t-1}$$
$$+ \beta_5 \text{PESA}_{i,t-1} + \mathbf{X}_{i,t-1}^{\text{attitudes}}\beta_6 + \mathbf{X}_{i,t}^{\text{behavior}}\beta_7 + \varepsilon_{i,t} \tag{2}$$

The demographic, attitudinal, and behavioral measures that enter the model are identical to those used in our models for introductory students. $\text{IESA}_{i,t-1}$ and $\text{IESA}_{i,t}$ represent the standardized scores on the intermediate microeconomics

assessment taken at the end of the course and at follow-up, respectively. $MESA_{i,t-1}$ and $PESA_{i,t-1}$ are scores on the two prerequisite skill assessments taken at the beginning of the term.

## Data description

Tables 2, 3, and 4 describe our analysis variables for our three cohorts of introductory microeconomics students. The first column of each table provides means of the variables that were collected at baseline, while the second and third columns provide means for these variables for the subsets of students that were successfully followed as well as for the variables that were collected at follow-up.

Our follow-up success rates ranged between 25% and 46%, but it is important to consider how our data collection strategy may have biased our samples. We successfully followed up with about 10% of our target samples by surveying students in other popular economics courses, resulting in samples that likely overrepresent students who took additional economics courses. In our second stage of follow-up, we encouraged participants by offering gift cards and explaining the value of educational research. This suggests our samples may be disproportionately composed of students who are motivated either by financial incentives or by an intrinsic interest in contributing to research. Despite these potential biases, we find that all of our follow-up samples have similar distributions of observable characteristics when compared to students in the target sample that did not participate.

We can also see in Tables 2, 3, and 4 that PESA scores on average declined between the end of the course and both follow-ups for all three cohorts, though students retained most of what they knew, and declines were modest. In the Fall 2022 cohort there was only a 1-point decline by the end of the first year, while scores decline by 6 points at the one year follow up for the Fall 2019 cohort and by 3 points for the Fall 2020 cohort. At the two-year follow-up, declines from baseline were almost the same as those observed at the one year follow-up.

While only 14-16% of students reported having a job where they used economics in the year after the course, this increases to 37-42% in the two years after the course ended. Shares of students who either completed an economics course in the interim period or were currently enrolled are much higher, ranging from 65% to 82% depending on the cohort and year of follow-up.

Tables 5, 6, and 7 describe the target sample and the follow-up analysis samples for the intermediate microeconomics courses taught in Spring 2019, Fall 2019, and Fall 2020. The 1.5-year follow-up sample for the Spring 2019 cohort does not substantially differ on any of the observed characteristics from the students who were not followed, but the 2.5-year follow-up sample is somewhat more positively selected as their average baseline PESA score is significantly higher, and these students are more likely to find economics interesting and think about it daily than students who were not successfully followed. We see a similar pattern in the Fall 2019 cohort where the follow-ups are more likely to major in economics (significant for 1.5 year follow-up), have higher average baseline test scores, and have more positive attitudes toward economics. The 1.5 and 2.5-year follow-ups of the Fall 2020 are also somewhat more positively selected as they have significantly higher baseline PESA and MESA scores. The 2.5-year follow-up somewhat over-represents students who were sophomores when they took the course relative to juniors.

We observe declines in scores on end-of-term tests during the follow-up period that were somewhat larger than we observed for the introductory courses, though there is still substantial retention. The average decline for the Spring 2019 cohort was 7 points at 1.5 years and 11 points at 2.5 years, and the declines observed for the Fall 2019 cohort were 11 points at 1.5 years and 9 points at 2.5 years. Scores of students in the Fall 2020 cohort declined by 7 points and 8 points at the 1.5-year and 2.5-year follow-ups respectively.

Intermediate-level students were much more likely to report that they held jobs where they applied their economic skills than the introductory-level students (e.g., 24-26% at first follow-up for intermediate microeconomics versus aforementioned 14-16% for introductory microeconomics students). This is in part due to the longer follow-up periods, but there are likely to be more employers willing to higher more advanced students too.

**Table 5.** Intermediate microeconomics samples: Spring 2019.

| | Target Sample | Followed in | |
|---|---|---|---|
| | | **Fall 2020** | **Fall 2021** |
| IESA (%) | 56 | 56 | 60 |
| IESA at follow-up (%) | | 49 | 49 |
| Freshman | 0.39 | 0.38 | 0.47 |
| Sophomore | 0.38 | 0.45 | 0.33 |
| Junior | 0.13 | 0** | 0+ |
| Senior | 0.10 | 0.17+ | 0.20 |
| Female | 0.43 | 0.52 | 0.53 |
| URM | 0.15 | 0.10 | 0.13 |
| First Gen | 0.03 | 0.03 | 0.07 |
| Econ Major | 0.67 | 0.76 | 0.87+ |
| MESA Intermediate (%) | 82 | 83 | 83 |
| PESA (%) | 56 | 59+ | 63* |
| Found Economics Interesting | 0.52 | 0.52 | 0.80* |
| Thought about Economics Daily | 0.34 | 0.38 | 0.60* |
| Job Used Economics | | 0.24 | 0.67 |
| Econ Course (interim or current) | | 0.79 | 0.80 |
| Econ Course (interim) | | 0.62 | 0.73 |
| Econ Course (current) | | 0.55 | 0.47 |
| N | 61 | 29 | 15 |
| Response Rate | | 0.48 | 0.25 |

Symbols represent tests of equality of means for those students who were successfully followed
and those that were not: + $p < 0.1$, * $p < 0.05$, ** $p < 0.01$.

**Table 6.** Intermediate microeconomics samples: Fall 2019.

| | Target Sample | Followed in | |
|---|---|---|---|
| | | **Spring 2021** | **Spring 2022** |
| IESA (%) | 52 | 56* | 56+ |
| IESA at follow-up (%) | | 45 | 47 |
| Freshman | 0.07 | 0.11 | 0.11 |
| Sophomore | 0.60 | 0.62 | 0.62 |
| Junior | 0.26 | 0.15* | 0.19 |
| Senior | 0.07 | 0.11 | 0.08 |
| Female | 0.32 | 0.36 | 0.30 |
| URM | 0.13 | 0.09 | 0.08 |
| First Gen | 0.10 | 0.11 | 0.08 |
| Econ Major | 0.60 | 0.72* | 0.70 |
| MESA Intermediate (%) | 78 | 80 | 83+ |
| PESA (%) | 50 | 51 | 56** |
| Found Economics Interesting | 0.41 | 0.51+ | 0.43 |
| Thought about Economics Daily | 0.29 | 0.34 | 0.41+ |
| Job Used Economics | | 0.25 | 0.27 |
| Econ Course (interim or current) | | 0.81 | 0.59 |
| Econ Course (interim) | | 0.79 | 0.54 |
| Econ Course (current) | | 0.57 | 0.35 |
| N | 140 | 53 | 37 |
| Response Rate | | 0.38 | 0.26 |

Symbols represent tests of equality of means for those students who were successfully followed
and those that were not: + $p < 0.1$, * $p < 0.05$, ** $p < 0.01$.

**Table 7**. Intermediate microeconomics samples: Fall 2020.

| | Target Sample | Followed in | |
| --- | --- | --- | --- |
| | | Spring 2022 | Spring 2023 |
| IESA (%) | 50 | 52 | 57** |
| IESA at follow-up (%) | | 45 | 49 |
| Freshman | 0.04 | 0.07 | 0.03 |
| Sophomore | 0.53 | 0.59 | 0.84** |
| Junior | 0.36 | 0.30 | 0.08** |
| Senior | 0.07 | 0.04 | 0.05 |
| Female | 0.29 | 0.28 | 0.38 |
| URM | 0.14 | 0.13 | 0.16 |
| First Gen | 0.08 | 0.11 | 0.14+ |
| Econ Major | 0.66 | 0.69 | 0.68 |
| MESA Intermediate (%) | 80 | 85* | 85+ |
| PESA (%) | 54 | 56+ | 60** |
| Found Economics Interesting | 0.55 | 0.56 | 0.68+ |
| Thought about Economics Daily | 0.38 | 0.41 | 0.46 |
| Job Used Economics | | 0.26 | 0.38 |
| Econ Course (interim or current) | | 0.78 | 0.76 |
| Econ Course (interim) | | 0.69 | 0.76 |
| Econ Course (current) | | 0.57 | 0.49 |
| N | 120 | 54 | 37 |
| Response Rate | | 0.45 | 0.31 |

Symbols represent tests of equality of means for those students who were successfully followed and those that were not: $^+$ $p < 0.1$, $^*$ $p < 0.05$, $^{**}$ $p < 0.01$.

## Results

### The determinants of introductory microeconomics skills retention

Tables 8 and 9 contain our estimates of the regression model shown in Eq 1 that explain follow-up scores (1-year and 2-year respectively) for introductory microeconomics students.

In column (1) of Table 8, the model includes as regressors student baseline PESA scores and dummy variables for college year, omitting the freshman reference category. We find that higher scores at baseline yield significantly higher scores at follow-up, but the return is less than one-to-one. In other words, we see some regression to the mean: If a student performs well on the baseline test, but part of this performance is driven by successful "educated guesses" or choices based on knowledge which was not well-consolidated, these gains will not be preserved. The estimated intercept (-0.23) is the expected standardized score at follow-up, holding the initial assessment at its zero mean. As expected, the decline is significant and negative. Seniors seem to retain substantially more skills than first year students.

The second column adds several demographic measures, and we find that performance of female students declines by a statistically significant almost half a standard deviation relative to males. We see no significant differences due to any other demographic characteristics. We add a measure of students' math skills at the beginning of the course in the third column and find it has a moderate positive effect on how much knowledge is retained. Incorporating this control induces no real change in the magnitude or significance of the other coefficients. In the fourth column we add controls for student attitudes toward economics, and neither is a statistically significant predictor of skills at the follow-up.

In the final two columns, we include measures of self-reported interim behavior, and find that neither holding a job that applies economics nor completing a class in economics during the interim have any significant effect on retention of economic knowledge. Being enrolled in an economics course at the time of follow-up, however, has a strongly significant positive (0.30 SD) effect on scores at follow-up.

**Table 8. Introductory microeconomics: One-year follow-up.**

| | (1) | (2) | (3) | (4) | (5) | (6) |
|---|---|---|---|---|---|---|
| PESA (standardized) | 0.57** | 0.53** | 0.49** | 0.48** | 0.46** | 0.45** |
| | (0.061) | (0.061) | (0.064) | (0.064) | (0.066) | (0.065) |
| Sophomore | −0.14 | −0.073 | −0.077 | −0.077 | −0.029 | −0.019 |
| | (0.13) | (0.13) | (0.13) | (0.13) | (0.13) | (0.13) |
| Junior | 0.051 | 0.15 | 0.13 | 0.16 | 0.21 | 0.23 |
| | (0.17) | (0.17) | (0.16) | (0.16) | (0.17) | (0.17) |
| Senior | 0.56+ | 0.54* | 0.56* | 0.52* | 0.54+ | 0.69* |
| | (0.34) | (0.25) | (0.24) | (0.23) | (0.29) | (0.28) |
| Female | | −0.43** | −0.41** | −0.41** | −0.38** | −0.36** |
| | | (0.11) | (0.11) | (0.11) | (0.11) | (0.11) |
| URM | | −0.0022 | 0.035 | 0.018 | −0.0097 | −0.033 |
| | | (0.11) | (0.11) | (0.11) | (0.12) | (0.12) |
| First Gen | | −0.020 | −0.044 | −0.036 | −0.034 | −0.051 |
| | | (0.12) | (0.12) | (0.12) | (0.12) | (0.12) |
| Econ Major | | 0.20 | 0.18 | 0.15 | 0.12 | 0.080 |
| | | (0.12) | (0.12) | (0.13) | (0.13) | (0.13) |
| MESA Foundations (standardized) | | | 0.12+ | 0.11+ | 0.12+ | 0.13+ |
| | | | (0.067) | (0.067) | (0.069) | (0.069) |
| Found Economics Interesting | | | | 0.17 | 0.17 | 0.16 |
| | | | | (0.13) | (0.13) | (0.13) |
| Thought of Economics Daily | | | | −0.072 | −0.057 | −0.045 |
| | | | | (0.12) | (0.12) | (0.12) |
| Job used Economics | | | | | 0.024 | −0.011 |
| | | | | | (0.15) | (0.15) |
| Econ Course (interim or current) | | | | | 0.26+ | |
| | | | | | (0.13) | |
| Econ Course (interim) | | | | | | 0.15 |
| | | | | | | (0.12) |
| Econ Course (current) | | | | | | 0.30* |
| | | | | | | (0.12) |
| Constant | −0.23* | −0.033 | −0.040 | −0.081 | −0.28 | −0.32+ |
| | (0.10) | (0.14) | (0.14) | (0.14) | (0.17) | (0.17) |
| Observations | 248 | 248 | 248 | 248 | 248 | 248 |
| $R^2$ | 0.331 | 0.386 | 0.395 | 0.399 | 0.411 | 0.429 |

Standard errors in parentheses.
Dependent variable is standardized PESA score at follow-up.
Course-year fixed effects are included but not shown.
+ $p < 0.1$, * $p < 0.05$, ** $p < 0.01$.

The story changes somewhat in our analysis of the 2-year follow-up scores shown in Table 9. We now see that the positive relationship between initial math skills and knowledge retention has become more statistically significant. Students who are either enrolled in another economics course at the time of follow-up or completed one before the follow-up are retain significantly more information than those who are not, but we can no longer determine which type of class has a stronger effect.

## The determinants of intermediate microeconomics skills retention

Our findings for the intermediate-level course are in many ways similar to what we find in our analysis of introductory students. Focusing first on the 1.5-year follow-up analysis shown in Table 10, we see that female students score lower at the 1.5-year follow-up than male students, though the effect is not statistically significant. On the other hand, we see that URM students retain marginally significantly less than non-URM students.

**Table 9. Introductory microeconomics: Two-year follow-up.**

| | (1) | (2) | (3) | (4) | (5) | (6) |
|---|---|---|---|---|---|---|
| PESA (standardized) | 0.58** | 0.53** | 0.46** | 0.45** | 0.44** | 0.46** |
| | (0.078) | (0.076) | (0.082) | (0.082) | (0.083) | (0.081) |
| Sophomore | −0.11 | −0.085 | −0.076 | −0.050 | 0.042 | 0.020 |
| | (0.15) | (0.16) | (0.16) | (0.16) | (0.15) | (0.15) |
| Junior | −0.026 | 0.023 | −0.030 | 0.050 | 0.12 | 0.18 |
| | (0.18) | (0.18) | (0.18) | (0.18) | (0.17) | (0.18) |
| Senior | 0.066 | 0.055 | 0.17 | 0.15 | 0.48 | 0.36 |
| | (0.54) | (0.58) | (0.43) | (0.42) | (0.42) | (0.44) |
| Female | | −0.21 | −0.23+ | −0.22+ | −0.22+ | −0.23+ |
| | | (0.13) | (0.13) | (0.12) | (0.13) | (0.13) |
| URM | | −0.092 | −0.077 | −0.084 | −0.053 | −0.066 |
| | | (0.20) | (0.21) | (0.21) | (0.21) | (0.21) |
| First Gen | | 0.022 | −0.0076 | −0.0046 | −0.034 | 0.014 |
| | | (0.16) | (0.16) | (0.16) | (0.17) | (0.17) |
| Econ Major | | 0.21 | 0.18 | 0.11 | 0.10 | 0.082 |
| | | (0.19) | (0.18) | (0.18) | (0.19) | (0.19) |
| MESA Foundations (standardized) | | | 0.17* | 0.17* | 0.18** | 0.16* |
| | | | (0.068) | (0.065) | (0.067) | (0.065) |
| Found Economics Interesting | | | | 0.22 | 0.18 | 0.21 |
| | | | | (0.15) | (0.15) | (0.15) |
| Thought of Economics Daily | | | | 0.16 | 0.18 | 0.18 |
| | | | | (0.15) | (0.15) | (0.15) |
| Job used Economics | | | | | −0.072 | −0.060 |
| | | | | | (0.13) | (0.13) |
| Econ Course (interim or current) | | | | | 0.34* | |
| | | | | | (0.14) | |
| Econ Course (interim) | | | | | | 0.077 |
| | | | | | | (0.14) |
| Econ Course (current) | | | | | | 0.22 |
| | | | | | | (0.14) |
| Constant | −0.26* | −0.17 | −0.13 | −0.31+ | −0.53** | −0.45* |
| | (0.12) | (0.17) | (0.17) | (0.17) | (0.19) | (0.19) |
| Observations | 191 | 191 | 191 | 191 | 191 | 191 |
| $R^2$ | 0.350 | 0.368 | 0.389 | 0.411 | 0.430 | 0.424 |

Standard errors in parentheses.
Dependent variable is standardized PESA score at follow-up.
Course-year fixed effects are included but not shown.
+ $p < 0.1$, * $p < 0.05$, ** $p < 0.01$.

Students with higher prerequisite skills (math and economics) retain significantly more than students who entered the course less prepared, while attitudes toward economics continue to have little measurable effect. Students that held jobs during the interim period scored significantly higher than those who did not, but none of the course behavior variables are significant at the 1.5 year follow-up.

Our sample size at the 2.5-year follow up is quite small (89 observations), so it is not surprising that many of the findings from the earlier follow-up are no longer statistically significant, though signs and magnitudes of effects are similar, as can be seen in Table 11.

## Discussion

We have found several results that warrant consideration of potential underlying mechanisms. Although our current evidence does not conclusively identify these mechanisms, we propose several possibilities in the hope that future research will clarify the processes at work.

**Table 10**. Intermediate microeconomics: 1.5 year follow-up.

| | (1) | (2) | (3) | (4) | (5) | (6) |
|---|---|---|---|---|---|---|
| IESA (standardized) | 0.61** | 0.59** | 0.44** | 0.43** | 0.44** | 0.43** |
| | (0.090) | (0.091) | (0.092) | (0.092) | (0.092) | (0.093) |
| Freshman | −0.19 | −0.20 | −0.40 | −0.38 | −0.32 | −0.34 |
| | (0.28) | (0.29) | (0.28) | (0.28) | (0.30) | (0.30) |
| Junior | 0.028 | −0.025 | 0.030 | 0.040 | 0.015 | 0.030 |
| | (0.19) | (0.19) | (0.19) | (0.18) | (0.18) | (0.19) |
| Senior | −0.49$^+$ | −0.47 | −0.38 | −0.31 | −0.25 | −0.24 |
| | (0.27) | (0.29) | (0.26) | (0.28) | (0.31) | (0.31) |
| Female | | −0.21 | −0.11 | −0.080 | −0.098 | −0.092 |
| | | (0.16) | (0.16) | (0.16) | (0.16) | (0.16) |
| URM | | −0.45* | −0.36$^+$ | −0.40* | −0.42* | −0.40$^+$ |
| | | (0.19) | (0.19) | (0.20) | (0.20) | (0.21) |
| First Gen | | 0.064 | 0.051 | −0.017 | 0.035 | 0.012 |
| | | (0.24) | (0.22) | (0.23) | (0.24) | (0.24) |
| Econ Major | | −0.036 | 0.091 | 0.024 | −0.024 | −0.035 |
| | | (0.18) | (0.17) | (0.18) | (0.17) | (0.18) |
| MESA Intermediate (standardized) | | | 0.19* | 0.15$^+$ | 0.14 | 0.14 |
| | | | (0.095) | (0.089) | (0.089) | (0.090) |
| PESA (standardized) | | | 0.24* | 0.24* | 0.23* | 0.23* |
| | | | (0.12) | (0.11) | (0.11) | (0.11) |
| Found Economics Interesting | | | | 0.16 | 0.19 | 0.19 |
| | | | | (0.17) | (0.18) | (0.19) |
| Thought of Economics Daily | | | | 0.25 | 0.21 | 0.21 |
| | | | | (0.15) | (0.15) | (0.14) |
| Job used Economics | | | | | 0.32* | 0.30* |
| | | | | | (0.14) | (0.14) |
| Econ Course (interim or current) | | | | | 0.014 | |
| | | | | | (0.22) | |
| Econ Course (interim) | | | | | | −0.089 |
| | | | | | | (0.18) |
| Econ Course (current) | | | | | | 0.074 |
| | | | | | | (0.18) |
| Constant | −0.52* | −0.33 | −0.53* | −0.66* | −0.74* | −0.70* |
| | (0.21) | (0.29) | (0.26) | (0.26) | (0.34) | (0.31) |
| Observations | 136 | 136 | 136 | 136 | 136 | 136 |
| $R^2$ | 0.282 | 0.309 | 0.394 | 0.415 | 0.431 | 0.433 |

Standard errors in parentheses.
Dependent variable is standardized IESA score at follow-up.
Course-year fixed effects are included but not shown.
$^+$ $p < 0.1$, * $p < 0.05$, ** $p < 0.01$.

The first result is the most straight-forward: Performance on the assessment declined significantly between the initial assessment and the follow-ups in most of our specifications. While the first follow-up for the introductory students was sooner (1 year) than it was for the intermediate students (1.5 years), this cannot explain the difference in declines as the intermediate students lost more skills in 1.5 years than the introductory students lost in 2 years. We believe this difference is likely due to differences in the learning goals of the two courses. The intermediate-level material was more technical and thus potentially more difficult to retain, and there was more reinforcement of the introductory material after the fact than the intermediate material. In future work, we plan to investigate whether retention differed based on types of questions, categorizing them by sophistication and difficulty.

Second, we find that higher scores on the course prerequisite skill assessments are strongly positively associated with increased retention of economics skills in both the introductory and intermediate-level courses. We believe the likely mechanism is that these students were able to understand the course concepts at a deeper level than those who did not

**Table 11**. Intermediate microeconomics: 2.5 year follow-up.

| | (1) | (2) | (3) | (4) | (5) | (6) |
|---|---|---|---|---|---|---|
| IESA (standardized) | 0.64** | 0.65** | 0.63** | 0.64** | 0.66** | 0.66** |
| | (0.073) | (0.086) | (0.11) | (0.11) | (0.11) | (0.12) |
| Freshman | 0.032 | −0.066 | −0.056 | −0.043 | −0.013 | 0.010 |
| | (0.32) | (0.35) | (0.34) | (0.35) | (0.35) | (0.39) |
| Junior | −0.25 | −0.24 | −0.21 | −0.19 | −0.37 | −0.27 |
| | (0.24) | (0.27) | (0.27) | (0.27) | (0.36) | (0.37) |
| Senior | −0.64* | −0.74* | −0.66+ | −0.57 | −0.75+ | −0.65 |
| | (0.31) | (0.30) | (0.33) | (0.36) | (0.40) | (0.43) |
| Female | | −0.20 | −0.19 | −0.13 | −0.12 | −0.11 |
| | | (0.17) | (0.17) | (0.18) | (0.18) | (0.18) |
| URM | | 0.071 | 0.083 | 0.14 | 0.15 | 0.16 |
| | | (0.26) | (0.28) | (0.30) | (0.32) | (0.32) |
| First Gen | | −0.031 | −0.023 | −0.069 | −0.033 | −0.034 |
| | | (0.30) | (0.30) | (0.31) | (0.35) | (0.34) |
| Econ Major | | −0.24 | −0.20 | −0.22 | −0.17 | −0.19 |
| | | (0.18) | (0.18) | (0.18) | (0.19) | (0.19) |
| MESA Intermediate (standardized) | | | −0.11 | −0.12 | −0.14 | −0.12 |
| | | | (0.14) | (0.13) | (0.14) | (0.14) |
| PESA (standardized) | | | 0.13 | 0.13 | 0.14 | 0.12 |
| | | | (0.16) | (0.15) | (0.15) | (0.15) |
| Found Economics Interesting | | | | 0.034 | 0.053 | 0.033 |
| | | | | (0.18) | (0.19) | (0.19) |
| Thought of Economics Daily | | | | 0.24 | 0.22 | 0.23 |
| | | | | (0.19) | (0.19) | (0.19) |
| Job used Economics | | | | | 0.21 | 0.20 |
| | | | | | (0.21) | (0.21) |
| Econ Course (interim or current) | | | | | −0.27 | |
| | | | | | (0.28) | |
| Econ Course (interim) | | | | | | −0.11 |
| | | | | | | (0.29) |
| Econ Course (current) | | | | | | −0.068 |
| | | | | | | (0.21) |
| Constant | −0.65** | −0.28 | −0.40 | −0.62 | −0.60 | −0.70 |
| | (0.24) | (0.35) | (0.37) | (0.41) | (0.46) | (0.53) |
| Observations | 89 | 89 | 89 | 89 | 89 | 89 |
| $R^2$ | 0.424 | 0.443 | 0.456 | 0.470 | 0.481 | 0.478 |

Standard errors in parentheses.
Dependent variable is standardized IESA score at follow-up.
Course-year fixed effects are included but not shown.
+ $p < 0.1$, * $p < 0.05$, ** $p < 0.01$.

have a strong economic or mathematical background. This deeper level of understanding may have been more persistent than the more surface-level understanding that other students had at the end of the term.

Third, we see the roles of future economics courses and jobs that apply economics are quite different for students in the introductory and intermediate courses. Later economics courses seem to help students retain information from the introductory course in both follow-ups, and this is consistent with subsequent economics courses reinforcing what was learned in the original course. Holding a job where they apply economic concepts is not significantly associated with retention. Conversely, courses do not help retention of intermediate-level skills, but we believe few "advanced" under-graduate economics courses substantially apply or build on the technical skills that are evaluated in the assessment. Students who have jobs during the interim period do in fact retain more skills, but this may in part be due to selection.

That is, it is possible that students with strong skills are more successful finding jobs where they can apply the economics knowledge. On the other hand, it is possible that working on economics-related topics on the job solidified student understanding.

## Conclusion

In this paper, we have examined the determinants of learning retention for introductory and intermediate-level microeconomics skills. Our institution regularly administers standard learning assessments at the end of our introductory and intermediate courses, and we have supplemented this data by following up with students in these courses one to 2.5 years after the original courses have ended. During the follow-up surveys, we gave these students the same assessments and asked them about their interim academic and job-related activities. Demographic data was collected as a part of end-of-semester surveys in each course.

Our analysis shows that initial performance levels, some student demographic characteristics, and interim behavior all have significant effects on how much of what was learned in the original course is retained over time. These results have practical implications for how we teach economics given that our primary goal as instructors is to impart useful skills that stick with students over the long-term.

The most clearly applicable lesson learned is that students who come in with better preparation, especially in math, retain more economic skills than other students. These math skills are also major determinants of learning during the course implying that whatever we do to support students and build these skills before or during the course should yield short run and long run dividends.

We also find that students who take additional economics courses on average retain more of what they learned in the introductory course. Most instructors already encourage students take additional courses in their field, but this gives us even more reason to continue to do so. At the intermediate level, additional courses do not have a positive impact on retention. The reason may in part be that these courses build on economic concepts but tend not use the technical models emphasized in our intermediate microeconomic theory courses. We believe there needs to be better synchronization of the intermediate-level course and the more advanced classes.

Finally, we find that applying and building on economics skills in jobs may help students retain skills learned in intermediate microeconomics. If true, we should invest more in supporting students during their searches for economics-related summer internships as well as starting their careers after graduation.

We acknowledge that there are some limitations of our current work. Educational processes are inherently complex with a myriad of decisions made by instructors and students based on a range of factors that are both observed and unobserved. The measures we use in this paper are by their nature imperfect: Test scores are incomplete measures of student knowledge, application of economic skills in future courses and jobs is subjective, and our measure of knowledge retention does not distinguish between students remembering concepts taught during the term and re-learning concepts after the term is over. Given these issues, the reader should not interpret our results as definitive. The same can be said of most educational research.

A more concrete potential issue relates to sample selection. Because not all students that we recruit to participate in the study chose to participate, the analysis sample may not be representative of the original course participants. That said, based on the statistical tests of demographic characteristics and exam scores discussed above, we believe our samples reasonably represent the original population of students.

A related concern is external validity. Our analysis sample is determined by the data collection design and is limited to data from six terms of two courses taught by two instructors at one institution. While Cornell University is not representative of all universities, most economics departments around the world offer courses with content that is very similar to these courses. In addition, Cornell is a selective R1 university with a student body very much like other selective R1 universities in the US. Future research could replicate this work at other institutions, with other instructors, in other courses,

or even in other disciplines to determine whether long-term retention of knowledge and skills operates similarly in different contexts.

In the current paper, we highlight the findings of differences in knowledge retention across gender and minority status and leave to future work a deeper exploration of the mechanisms underlying the fact that, for example, female students seem to demonstrate lower material retention compared to their male counterparts. Another important aspect of differences in retention across demographic lines is their potential intersectionality. For example, there may be mechanisms that are specific to female students who come from under-served student populations. Given the size of our final analysis sample, we are unable to explore differences in retention in subgroups defined by gender and race, but we hope that future research using larger data sets will shed light on these issues.

In our own future research, we plan to supplement our intermediate microeconomics student data with retrospective surveys on study behavior that were collected during the course and at the time of the final exam. This should help us understand how differences in study behavior might affect retention of information. Moreover, we intend to examine retention on a question-by-question level, separately analyzing the determinants of retention, loss of knowledge, and improved performance compared to the original end-of-semester assessment. We will also plan to analyze assessment scores based on subsets of questions to see how retention of different types of skills varies across students. For example, it may be that students retain conceptual understanding better than their ability to solve more technical problems. Finally, we plan to analyze the role that modality of delivery (online vs. in-person) and pedagogical approach (pure lecture vs. use of active learning methods) have on student knowledge retention over time.

Perhaps the most important contribution of our paper is the process we have developed for following undergraduate students in the years after they complete courses. While this requires significant time and resources, we have shown it is still quite feasible to collect this kind of data and obtain reasonable response rates and representative samples. We hope other researchers will learn from our experience and go on to collect their own data to learn about long-term learning in other contexts.

## Acknowledgments

This study was determined to be exempt from review by Cornell University's Institutional Review Board (Protocol No. 1708007347).

The authors would like to thank Nahid Hassan and Sarah Almosawari for their research assistance, and Carolyn Aslan and Lisa Sanfillipo from the Cornell Center for Teaching Innovation for their support during the IRB process. We are grateful to Peter Lepage for founding the Active Learning Initiative at Cornell and believing in our vision. Third, we thank all of the students who participated in this study by taking numerous assessments and surveys for their trust and hard work. Last but not least, we thank Robin Taylor (RTRES Consulting) for her outstanding contributions as the project's external evaluator.

## Author contributions

**Conceptualization:** Douglas McKee, George Orlov.

**Data curation:** Douglas McKee, George Orlov.

**Formal analysis:** Douglas McKee, George Orlov.

**Funding acquisition:** Douglas McKee, George Orlov.

**Investigation:** Douglas McKee, George Orlov.

**Methodology:** Douglas McKee, George Orlov.

**Project administration:** Douglas McKee, George Orlov.

**Resources:** Douglas McKee.

**Software:** Douglas McKee.

**Supervision:** Douglas McKee.

**Validation:** Douglas McKee.

**Visualization:** Douglas McKee, George Orlov.

**Writing – original draft:** Douglas McKee, George Orlov.

**Writing – review & editing:** Douglas McKee, George Orlov.

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
