## [Decision Letter · Decision Letter 0]

1 Apr 2025

PONE-D-25-05136As Time Goes By: Long-term Retention of Economics SkillsPLOS ONE

Dear Dr. McKee,

Thank you for submitting your manuscript to PLOS ONE. After careful consideration, we feel that it has merit but does not fully meet PLOS ONE’s publication criteria as it currently stands. Therefore, we invite you to submit a revised version of the manuscript that addresses the points raised during the review process. Both reviewers believed that work on this issue is valuable, but have several comments that you will either need to address or refute in your response.  In addition, you will need to discuss more carefully your sample and any factors that might affect the external validity of your findings.  For example, you identify students for follow-up surveys based on students in courses with multiple students who participated in the initial assessment.  Are all these classes economics courses?  If not, please clarify.  If so, please discuss how this may bias your results.  Furthermore, there is little information on how the courses are taught and what is emphasized in the courses.  Are multiple choice exams used in the courses? Do the courses stress the use of graphs and equations or is the intro course a more issues-oriented course?  More broadly, how does the fact that there is only one instructor teaching these classes affect external validity?  Could this be construed as a study of the impact of one instructor's pedagogical choices?

We look forward to receiving your revised manuscript.

Kind regards,

Ann L. Owen

Academic Editor

PLOS ONE

Journal Requirements:

3. Thank you for stating the following financial disclosure: [This paper is based upon work supported by the National Science Foundation under Grant 2021094, and any opinions, findings, and conclusions or recommendations expressed in this paper are those of the authors and do not necessarily reflect the views of the National Science Foundation.]. 

Reviewers' comments:

Reviewer's Responses to Questions

**Comments to the Author**

1. Is the manuscript technically sound, and do the data support the conclusions?

Reviewer #1: Partly

Reviewer #2: Yes

2. Has the statistical analysis been performed appropriately and rigorously?

Reviewer #1: Yes

Reviewer #2: Yes

3. Have the authors made all data underlying the findings in their manuscript fully available?

Reviewer #1: No

Reviewer #2: Yes

4. Is the manuscript presented in an intelligible fashion and written in standard English?

Reviewer #1: Yes

Reviewer #2: Yes

5. Review Comments to the Author

Reviewer #1: 1. The literature review seems to not consider economics articles on retention. Juttler (2020, Feb) in this journal or Cosgrove and Olitsky (SEJ, 2015) which although for high school is still relevant.

2. The next to last paragraph says that a limitation of other studies is not considering the role of some characteristics. You say this is helpful. Expand on why.

3. Target sample is misleading. Isn't the entire class the target? The description of the samples was difficult to follow.

4. Do you know how much time the spent the first time the completed the assessment?

5. Better explain the normalization. what are the new mean and s.d.?

6. You say MESA scores declined substantially. Substantial is subjective. The percentage decline suggests a 1.8 item decline on average. My interpretation is that is substantial retention.

7. Is it retention? Suppose only a 3 item quiz and initially the student gets 2 of 3 correct. On the retake they get 1 of 3 correct, but the one they got correct is the one they initially got wrong. You would say they retained half of what the initially knew, but in fact they retained nothing.

8. Is IESA asking about MRS and MRTS? Should we expect students to remember that? In micro you learn of sunk costs, that can come up in life (same with supply and demand).

9. Do you have enough observations to report a regression for Spring 2023 of the intermediate sample?

10. The entire discussion section is your unsubstantiated opinion of why results look that way. For example, you have no idea why female students have different scores. I would argue for deleting this section.

Reviewer #2: The aim of the study is to capture what students retain in the very long run AND what determines that retention.

The paper provides evidence that retention of course skills is associated with some demographic characteristics and prior preparation for the course. It goes beyond existing studies on long-term learning by exploring such factors.

The study is well executed but there is a need to provide a more thorough explanation in some places of the limits of the measures used. More details are provided below.

Good context is provided 1) on some factors influencing short-term retention in economics and other disciplines, and 2) on long term learning from literature in disciplines other than economics, where such literature is sparse. The dearth of studies examining long-term learning in economics underscores the value of this study.

The samples used pool data from three cohorts for each course, giving a reasonable sample sizes for each of the courses (with fixed effects included for each cohort).

The study doesn’t raise any ethical concerns: it is reported as being IRB exempt, student consent was obtained in advance and data were de-identified.

P3: line 77-79: Be explicit about how the differences in course teaching method and modality are captured in the cohort fixed effect

P3: line 89 and P7: line 180: Can you say more about why the reader should be confident that 30 (or 31 for intermediate) multiple choice questions are a reasonable measure of student learning?

P3: line 99 (and equivalent for intermediate): What can you say about the external validity of the findings (beyond your own institution)?

P6: lines 124-126: Provide further explanation about how information on “courses that used their economic knowledge” and jobs where “they applied their economic knowledge” was gathered. Was this based on the student’s own interpretation of what using/applying economic knowledge meant? Was economic knowledge defined, or confined specifically to microeconomic knowledge? The specifics of the additional course might matter – e.g. upper level applied micro course.

The results are presented separately for intro and intermediate micro, implying that these samples are separate. Might some of the students in the (earlier) intro cohorts also be included in the intermediate sample? If so, does that/how might that matter?

P10: In discussing the results, how do you distinguish between retention of learning from intro micro and *re-learning* concepts in another economics course?

P13: Line 320 and 325: Do you have evidence to support the purported gender difference in study habits? If so, provide detail. If not, it would be better to remove purely speculative material.

P13: Line 342: This depends on what is being captured in the “jobs that apply economics concepts”. (See earlier comment about defining this more fully.)

Some of these questions are likely not answerable in a definitive way: the complexities in interpreting the results should be explicitly acknowledged.

The variables used are by their nature imperfect proxies. The measures used seem reasonable – but the paper should discuss the measurement complexities. (e.g. retention v relearning, what is “using economic knowledge” in a job, what about “using economic knowledge” in other parts of life etc.?)

The paper tackles a very difficult question – and adds to our knowledge.

6. PLOS authors have the option to publish the peer review history of their article (what does this mean?). If published, this will include your full peer review and any attached files.

Reviewer #1: No

Reviewer #2: No

---

## [Author Response · Author response to Decision Letter 1]

11 Aug 2025

See attached response-to-reviewers.pdf

---

## [Decision Letter · Decision Letter 1]

11 Sep 2025

As Time Goes By: Long-term Retention of Economics Skills

PONE-D-25-05136R1

Dear Dr. McKee,

We’re pleased to inform you that your manuscript has been judged scientifically suitable for publication and will be formally accepted for publication once it meets all outstanding technical requirements.

Kind regards,

Ann L. Owen

Academic Editor

PLOS ONE

Additional Editor Comments (optional):

Reviewer #1:

Reviewer #2:

Reviewers' comments:

Reviewer's Responses to Questions

**Comments to the Author**

1. If the authors have adequately addressed your comments raised in a previous round of review and you feel that this manuscript is now acceptable for publication, you may indicate that here to bypass the “Comments to the Author” section, enter your conflict of interest statement in the “Confidential to Editor” section, and submit your "Accept" recommendation.

Reviewer #1: All comments have been addressed

Reviewer #2: All comments have been addressed

2. Is the manuscript technically sound, and do the data support the conclusions?

Reviewer #1: Yes

Reviewer #2: Yes

3. Has the statistical analysis been performed appropriately and rigorously?

Reviewer #1: Yes

Reviewer #2: Yes

4. Have the authors made all data underlying the findings in their manuscript fully available?

Reviewer #1: Yes

Reviewer #2: Yes

5. Is the manuscript presented in an intelligible fashion and written in standard English?

Reviewer #1: Yes

Reviewer #2: Yes

6. Review Comments to the Author

Reviewer #1: Thank you for addressing my concerns. Any remaining concerns are beyond the scope of this manuscript.

Reviewer #2: (No Response)

7. PLOS authors have the option to publish the peer review history of their article (what does this mean?). If published, this will include your full peer review and any attached files.

Reviewer #1: No

Reviewer #2: No

---

## [Editor Report · Acceptance letter]

PONE-D-25-05136R1

PLOS ONE

Dear Dr. McKee,

I'm pleased to inform you that your manuscript has been deemed suitable for publication in PLOS ONE. Congratulations! Your manuscript is now being handed over to our production team.

Kind regards,

on behalf of

Dr. Ann L. Owen

Academic Editor

PLOS ONE